# Investigation of the Kinetics and Reaction Mechanism for Photodegradation Tetracycline Antibiotics over Sulfur-Doped Bi_2_WO_6-x_/ZnIn_2_S_4_ Direct Z-Scheme Heterojunction

**DOI:** 10.3390/nano11082123

**Published:** 2021-08-20

**Authors:** Yanbo Jiang, Kai Huang, Wei Ling, Xiandong Wei, Yijing Wang, Jun Wang

**Affiliations:** 1Research Center of Wastewater Engineering Treatment & Resource Recovery, Guangxi Beitou Environmental Protection & Water Group Co., Ltd., Nanning 530029, China; yamboo@gxu.edu.cn (Y.J.); lingwei1991@yahoo.com (W.L.); weixiandong@yahoo.com (X.W.); 2National Engineering Research Center for Non-Food Biorefinery, Guangxi Key Laboratory of Bio-Refinery, Institute of Eco-Environmental Research, Guangxi Academy of Sciences, Nanning 530007, China; 3Institute of Ecological Engineering, Guangxi University, Nanning 530004, China; 4Department of Environment, School of Architectural Engineering, Guangxi University for Nationalities, Nanning 530006, China; jinnywang@gxun.edu.cn

**Keywords:** z-scheme, photocatalytic degradation, synergistic effect, S^2−^ doping, tetracycline hydrochloride, surface oxygen vacancies

## Abstract

The rational design of direct Z-scheme heterostructural photocatalysts using solar energy is promising for energy conversion and environmental remediation, which depends on the precise regulation of redox active sites, rapid spatial separation and transport of photoexcited charge and a broad visible light response. The Bi_2_WO_6_ materials have been paid more and more attention because of their unique photochemical properties. In this study, S^2−^ doped Bi_2_WO_6-x_ coupled with twin crystal ZnIn_2_S_4_ nanosheets (Sov−BWO/T−ZIS) were prepared as an efficient photocatalyst by a simple hydrothermal method for the removal of tetracycline hydrochloride (TCH). Multiple methods (XRD, TEM, XPS, EPR, UV vis DRS, PL etc.) were employed to systematically investigate the morphology, structure, composition and photochemical properties of the as-prepared samples. The XRD spectrum indicated that the S^2−^ ions were successfully doped into the Sov−BWO component. XPS spectra and photoelectrochemical analysis proved that S^2−^ served as electronic bridge and promoted captured electrons of surface oxygen vacancies transfer to the valence band of T−ZIS. Through both experimental and in situ electron paramagnetic resonance (in situ EPR) characterizations, a defined direct Z-scheme heterojunction in S-BWO/T−ZIS was confirmed. The improved photocatalytic capability of S-BWO/T−ZIS results ascribed that broadened wavelength range of light absorption, rapid separation and interfacial transport of photoexcited charge, precisely regulated redox centers by optimizing the interfacial transport mode. Particularly, the Sov−50BWO/T−ZIS Z-scheme heterojunction exhibited the highest photodegradation rate was 95% under visible light irradiation. Moreover, this heterojunction exhibited a robust adsorption and degradation capacity, providing a promising photocatalyst for an organic pollutant synergistic removal strategy.

## 1. Introduction

Organic pollutants in an aquatic ecosystem cause continuous environmental deterioration, which greatly threatens the ecological environment and human public health. Among various organic pollutants, the increasing presence of antibiotics and textile dyes in wastewater is one of the most serious issues. For instance, tetracycline hydrochloride (TCH), as one of the main components of pharmaceutical wastewater, is usually chemically stable and hardly removed by conventional biochemical and physical measures. It can bring about a series of environmental concerns such as causing the formation of resistant microorganisms and upsetting the natural balance [1,2,3,4]. To mitigate the threats and promote the sustainable and rapid development of industry, it is an emergency to exploit novel technologies aimed at the purification of industrial wastewater. Various wastewater treatment techniques (e.g., adsorption, membrane separation, coagulation, biodegradation and advanced oxidation processes (AOPs)) have been proven to be effective tactics for removing organic pollutants from wastewater. Among these strategies, adsorption, as one of the simplest and most efficient treatment methods, is strongly recommended for the extensive removal of organic pollutants [5]. However, its limited adsorption capacity, increased regeneration cost and potential secondary pollution limit its wide application. Therefore, it is necessary to find a feasible and environmentally friendly adsorbent regeneration technology.

Recently, photocatalysis as one kind of the most “green” AOPs, has emerged as a promising approach for the removal of TCH without bringing secondary pollution in the wastewater treatment process. Especially, various photocatalysts possess high specific surface area and ordered mesoporous layers, which facilitates the adsorption of contaminants within the mesoporous layer [6,7,8]. Moreover, it is beneficial to the subsequent photocatalytic degradation through creating a partial microenvironment of organics, thus better satisfying the requirements for highly efficient removal of pollutants. Besides, the active component (OH, O_2_^−^, h^+^, e^−^) produced by photocatalysts could immediately react with adsorbed contaminants, reviving the regenerate adsorption ability. Currently, bismuth tungstate (Bi_2_WO_6_), as the simplest Aurivillius-type member with alternating (Bi_2_O_2_)_n_(WO_4_)_n_ construction and simultaneously containing the WO_6_ perovskite component, has attracted intense attention in the environmental remediation of refractory organic wastewater [9,10,11,12]. Especially, surface vacancy-regulated Bi_2_WO_6_ (Bi_2_WO_6-x_) is widely used in photocatalytic purify of organic contaminants due to its appropriate bandgap structure and positive valence band level, visible-light response and robust photochemical stability. In previous reports, introducing heteroatom has proven to be one of the most effective modification strategies, because the band structure and electronic structure can be effectively adjusted, ultimately optimizing the morphology, improving photoelectrochemical properties and enhancing the photocatalytic performance [13].

To date, it is widely attracting more and more attention in Z-scheme heterojunction photocatalysts, due to their maximum redox capacity and effective separation and transfer of photoexcited carriers. However, the building of a non-intimate contact heterogeneous interface has been proven to be ineffective in the development of an extraordinary Z-scheme system [14,15,16]. In 2018, Hao et al. found that the induction of surface oxygen vacancies (Sov) in a Z-scheme system possessed several distinct advantages: on the one hand, the introduced Sov facilitated the formation of intimate contact. Simultaneously, it enhanced the separation and migration efficiency of photoexcited carriers due to its built-in electric field; on the other hand, Sov could effectively adjust the band structure to realize the optimized Sov-mediated Z-scheme systems [17]. Furthermore, Sun et al. investigated firstly that Sov-dominated “Electron Bridge” greatly enhanced the spatial separation ability and utilization efficiency of photoexcited carriers [18]. Therefore, considering the unique advantages of Sov in heterojunction, coupling Sov-mediated porous nanosheets with other porous nanosheets to build electron bridge in Z-scheme heterojunction is highly estimated to regulate the photocatalytic degradation efficiency, but has rarely been reported.

In this study, a series of S^2−^ doped Bi_2_WO_6-x_ coupling with twin crystal ZnIn_2_S_4_ (Sov−BWO/T−ZIS) composites were fabricated by self-assembly method via the tactics that synergistic adsorption with photocatalytic decontamination. Benefits from the S^2−^ doping and the electron trap consisted of Sov, the as-prepared Sov−BWO/T−ZIS Z-scheme heterojunctions displayed the rapid separation and efficient utilization of carriers, thus exhibiting the enhancement of TCH removal efficiency.

## 2. Materials and Methods

### 2.1. Materials

Zinc chloride (ZnCl_2_), indium chloride (InCl_3_·4H_2_O), thioacetamide (TAA), glycerol, sodium tungstate (Na_2_WO_4_·2H_2_O), bismuth nitrate (Bi(NO_3_)_3_·5H_2_O) and N, N-dimethylformamide (DMF) were obtained from Shanghai Macklin Biochemical Co., Ltd. (88 Darwin road, Pudong, Shanghai, China). All used chemicals were used as received without further purification.

### 2.2. Synthesis

#### 2.2.1. Preparation of Bi_2_WO_6-x_ Nanosheets with Surface Oxygen Vacancies (Sov−BWO)

The Bi_2_WO_6-x_ nanosheets were synthesized via the following steps: firstly, 485 mg Bi(NO_3_)_3_·5H_2_O was added into 80 mL deionized water for 20 min with the aid of magnetic stirring. Then, 165 mg Na_2_WO_4_·2H_2_O was added into the aqueous solution and sustained another 10 min. Then, the appeared precipitate was centrifuged several times and transferred into 35 mL deionized water. Then, the mixed solution was transferred into a Teflon-lined autoclave and kept at 433 k for 18 h. Finally, the products were washed several times with deionized water and dried at 333 k for 12 h. The as-prepared products were denoted as Sov−BWO.

#### 2.2.2. Preparation of Bi_2_WO_6-x_/ZnIn_2_S_4_ Composite Photocatalysts with Surface Oxygen Vacancies-Rich (Sov-xwt%BWO/T−ZIS)

The Bi_2_WO_6-x_/ZnIn_2_S_4_ composites were synthesized through a facile in situ solvothermal method, which was showed as Scheme 1. In a typical process, a certain amount of Bi_2_WO_6-x_ sample (25 mg, 50 mg, 75 mg, 100 mg) was dispersed into the 15 mL DMF solution. Then, ZnCl_2_ (1 mmol) was dissolved into DMF and maintained ultrasonic for 10 min. InCl_3_·4H_2_O (2 mmol) was added into the mixture solution and maintained 10 min. TAA (4 mmol) was dispersed in the mixture solution, which was going on ultrasonic processing for 5 min. Subsequently, glycerol (3 mL) was added into the mixture and maintained for 5 min. The collected solution was then transferred into a Teflon-lined autoclave (50 mL) and heated at 453 K for 10 h. Finally, after cooling down to room temperature, the as-prepared product was obtained and labeled as Sov−25BWO/T−ZIS, Sov−50BWO/T−ZIS, Sov−75BWO/T−ZIS, Sov−100BWO/T−ZIS. For comparison, the synthetic procedure of the pure ZnIn_2_S_4_ sample was similar to the above steps except that the Bi_2_WO_6-x_ was not added and labeled as T−ZIS.

### 2.3. Characterization

X-ray Diffraction (XRD) was obtained by a Rigaku D/MAX 2500 V (Matsubara-choAkishima-shi, Tokyo, Japan), a scanning scope from 10° to 80°. Fourier transform infrared spectroscopy (FT-IR) spectra of the samples were conducted using a Nicolet iS 50 (TMO, Waltham, MA, USA). The microstructures were monitored by transmission electron microscopy (TALOS F200 TEM, TMO, Waltham, MA, USA). The morphology of the samples was monitored on a field-emission scanning electron microscopy (S-4800 FE-SEM, HITACHI, Chiyoda-ku, Tokyo, Japan) equipped with an energy-dispersive X-ray spectrometry (EDS). The positions of the XPS peaks were determined with reference to the carbon peak of C ls at 284.8 eV. The binding energies of Zn, In, S, Bi, W and O elements were acquired using an X-ray photoelectron spectroscopy (Nexsa, XPS, TMO, Waltham, MA, USA). The electron paramagnetic resonance (EPR) analysis was performed to test surface oxygen vacancies on a Bruker A300 spectrometer at 298 K. In situ electron paramagnetic resonance (in situ EPR) spectra were obtained on a Bruker EMXplus spectrometer equipped with a 150 W mercury lamp as the illumination source. The details of the instrumental parameters were as follows: Scanning frequency: 9.82 GHz, central field: 3400 G, scanning power: 20 mW, and scanning temperature: 298 K (room temperature, RT). The optical absorption of samples was gotten by a UV-vis DRS spectrometer (UV-3600 Plus, SHIMADZU, Nakagyo-ku, Kyoto, Japan), in which BaSO_4_ acted as the internal reflectance standard. Photoluminescence (PL) spectroscopy was measured using a PerkinElmer LS55 photoluminescence spectrometry testing system under the excitation of 325 nm light. Liquid chromatography mass spectrometry (LC-MS) was measured on the Agilent 1290 UPLC/6540 Q-TOF (Santa Clara, CA, USA).

### 2.4. Adsorption and Photocatalytic Performance Test

Using tetracycline hydrochloride (TCH) as a model pollutant, the decontamination activity of as-prepared materials was evaluated. The initial pH of the TCH solution was 6.3. The photocatalyst (T−ZIS, Sov−BWO) was added into solution, increasing PH to 6.7. The adsorption capacity and photocatalytic activity of the as-prepared samples (T−ZIS, Sov−BWO, Sov−25BWO/T−ZIS, Sov−50BWO/T−ZIS, Sov−75BWO/T−ZIS and Sov−100BWO/T−ZIS) was investigated in the glass vial with circulating cooling water system. A 300 W Xe lamp with a cut-off filter (λ > 400 nm) was used as the source of visible light. Typically, 30 mg as-synthesized sample was dispersed in 100 mL TCH aqueous solution (20 mg/L), and stirred magnetically in the dark for 30 min to achieve adsorption-desorption equilibrium. Then, 2 mL suspension was withdrawn each 15 min and then filter through 0.22 μm nylon membrane. The TCH concentrations were measured using a UV–vis spectrophotometer (UV-5100, METASH, Shanghai, China) at the absorption peak of 357 nm. The stability and reusability of the best-performing sample were tested for five runs under the same conditions.

### 2.5. Active Species Trapping Experiments

Ethylenediaminetetraacetic acid disodium (EDTA-2Na), isopropanol, p-benzoquinone of 1 mmol·L^−1^ were acted as scavengers to capture hole (h^+^), hydroxyl radical (OH) and superoxide radical (O_2_^−^), respectively. The experiments were similar to the above best-performing photocatalytic degradation of TCH.

The adsorption and photocatalytic degradation rates were calculated by using the following equations:Removal efficiency = (C_0_ − Ct)/(Ct)(1)
where C_0_ was the absorbance of the original TCH solution. The Ct is the absorbance of the residual concentration of TCH under different degradation times.

### 2.6. Photoelectrochemical Measurements

A CHI660E electrochemical workstation was performed to test. In a typical three-electrode system, Pt foil acted as the counter electrode and Ag/AgCl served as a reference electrode, in which the electrolyte was 0.5 M Na_2_SO_4_ aqueous solution. Electrochemical Impedance Spectroscopy (EIS) was carried out in 0.5 M Na_2_SO_4_ aqueous solution over the frequency ranging from 0.1 Hz to 10^5^ Hz. Mott-Schottky plots were measured under the frequency of 1000 Hz and 2000 Hz in 0.5 M Na_2_SO_4_ aqueous solution. Photocurrent response (I-t) of the as-prepared samples was measured on IVIUM electrochemical workstation. A single wavelength (λ = 375 nm) was served as the light source.

## 3. Results and Discussion

### 3.1. Structure Characterizations

Appendix A showed XRD patterns of the as-prepared T−ZIS, Sov−BWO, Sov−25BWO/T−ZIS, Sov−50BWO/T−ZIS, Sov−75BWO/T−ZIS and Sov−100BWO/T−ZIS samples. The characteristic peaks at 28.3°, 32.7°, 47.1°, 55.9° and 58.5°of pristine Sov−BWO could be attributed to the (131), (200), (202), (133) and (262) crystal planes, corresponding to the Bi_2_WO_6_ (Orthorhombic, JCPDS 39–0256), respectively [18]. Meanwhile, the diffraction peaks at 20.6°, 27.5° and 47.5° were assigned to the (006), (102) and (110) crystal planes of ZnIn_2_S_4_ (Hexagonal, JCPDS 65–2023), respectively [19]. The peak at 13.9° could be attributed to the (111) plane of cubic ZnIn_2_S_4_ (JCPDS 48–1778) [20]. Notably, the XRD patterns of the Sov−BWO/T−ZIS Z-scheme heterojunctions were similar to that of T−ZIS, indicating that the Sov−BWO coupling had no adverse influence on the crystal structure of T−ZIS. However, it was clearly observed in Figure 1a that the (102) diffraction peaks of Sov−BWO/T−ZIS composites significantly shifted lower than that of T−ZIS. In addition, with the increase of Sov−BWO, the intensity of crystal plane (131) changed from weak to strong, which indicated that the introduction of T−ZIS reduced the self-stacking of Sov−BWO, promoted the cohesion of nanosheets and optimized the morphology. Meanwhile, the new peaks at 24.9° were observed in all heterojunctions and belonged to S^2−^ doping into Sov−BWO [21,22,23]. Furthermore, combined with XPS analysis, it was convincingly confirmed the existence of S^2−^ doping, which promoted the rapid transfer of photoexcited charge on the unique interface. The FT-IR spectra of the as-obtained samples were shown in Figure 1b. The characteristic peaks of pristine Sov−BWO at 581 cm^−1^ and 726 cm^−1^ were considered as W–O bonds, while the characteristic peak at 1382 cm^−1^ was attributed to a W–O–W vibration bond, respectively [24]. Appendix A showed that the characteristic peaks of the T−ZIS, Sov−BWO and S^2−^ doped Sov−BWO/T−ZIS samples were around 1619 cm^−1^ and 3421 cm^−1^, respectively, which were caused by O–H bonds and water molecules adsorbed on the surface [25].

The morphology of prepared Sov−BWO, T−ZIS and Sov−50BWO/T−ZIS samples were investigated by FE-SEM. As shown in Figure 1c, the pristine Sov−BWO sample demonstrates obvious irregular ultrathin nanosheets. Figure 1d provided the FE-SEM image of petal-like T−ZIS. As displayed in the SEM image from Figure 1e, the as-obtained Sov−50BWO/T−ZIS photocatalyst exhibited self-assembly nanosheets with a tight contact. It was clearly revealed that the introduction of Sov−BWO into T−ZIS would generate the cohesion of petal-like structures. In addition, the Sov−BWO nanosheets were uniformly assembled with T−ZIS nanosheets to establish two dimension (2D/2D) tight heterogeneous interfaces. Composition and distribution of elements in Sov−50BWO /T−ZIS hybrid was revealed by the EDS spectrum and elemental mapping images and shown in Appendix A. The EDS spectrum confirmed the presence of Zn, In, S, Bi, W and O elements in Sov−50BWO/T−ZIS samples. Moreover, the elements of Bi, W and O were evenly distributed, which was consistent with uniform anchoring and tight self- assembly.

Furthermore, HRTEM images of Sov−50BWO/T−ZIS sample were presented to further analyze the morphology. HRTEM image in Figure 1f, the crystal spacing was 0.19 nm and 0.32 nm, corresponding to the (202) plane of Bi_2_WO_6_ and the (102) plane of ZnIn_2_S_4_, respectively. The typical lattices of Sov−BWO and T−ZIS indicated the successful construction of the Sov−50BWO/T−ZIS hybrid. Furthermore, as shown in Figure 1g, the discontinuity of the lattice could be seen clearly, which arose from the existence of an oxygen vacancy in Sov−BWO [26]. More meaningfully, it was intuitively observed that T−ZIS had a (102) crystal plane symmetrical twin crystal structure, which was in accordance with previous literature (Figure 1f,g) [27]. Moreover, it shown that the false-color image in the (102) plane corresponded to twin crystal T−ZIS. Due to the unique advantage of building crystal structure, it facilitated an internal electric field in the twinned T−ZIS, thereby improving the separation efficiency of electrons and holes in the photocatalytic reaction of twin crystal T−ZIS. It revealed that twinned crystals can promote adsorption and increase photocatalytic reactions.

The surface chemical state of surface elements was analyzed by XPS. Figure 2a presented the XPS survey scan of as-prepared Sov−BWO, T−ZIS and Sov−50BWO/T−ZIS samples. It can be observed that Bi, W and O are the principal elements of Sov−BWO. In addition, T−ZIS consisted of Zn, In and S elements, while the Bi, W, O, Zn, In and S exist in the Sov−50BWO/T−ZIS sample. Figure 2b,c displayed high resolution spectra of Zn 2p and In 2d. For pure T−ZIS, the two distinct peaks at 1044.7 eV and 1021.7 eV are in accordance with Zn 2p_1/2_ and Zn 2p_3/2_, while the two peaks at 452.5 eV and 445.0 eV were distributed to In 3d_3/2_ and In 3d_5/2_, respectively. Similarly, as for Sov−50BWO/T−ZIS sample, the characteristic peaks at 1145.2 eV and 1122.1 eV are assigned to the Zn 2p_1/2_ and Zn 2p_3/2_ of Zn^2+^ species, while the peaks at 452.4 eV and 444.9 eV were consistent with the In 3d_3/2_ and In 3d_5/2_ of In^3+^, respectively [28]. These relatively larger shifts of Zn^2+^ and In^3+^ revealed the strong influence of T−ZIS, compared to the previous literature with heterojunction formation [29,30,31]. Meanwhile, it showed in Figure 2d that the S 2p (S 2p_1/2_ and S 2p_3/2_ at 162.9 and 161.7 eV) of Sov−50BWO/T−ZIS hybrid exhibited the positive shift compared to T−ZIS (S 2p_1/2_ and S 2p_3/2_ at 162.8 eV and 161.6 eV). Whereas the Bi 4f characteristic peaks at 163.6 eV and 158.2 eV of Sov−50BWO/T−ZIS shifted toward lower binding energy compared to the peaks at 164.4 and 159.0 eV of Sov−BWO. Interestingly, the obvious new peaks approximately at 164.3 eV and 158.9 eV were attributed to the reason that S^2−^ doped into Sov−BWO [32]. The strong changes revealed that the S^2−^ acted as an efficient electronic bridge and facilitated the trapped electron carriers transfer from Sov−BWO to T−ZIS. In Figure 2e, the O 1s of the original Sov−BWO was divided into two peaks at 530.8 eV and 529.8 eV, which were attributed to the widespread existence of lattice oxygen and the absorption of oxygen on the oxygen vacancies in the Sov−BWO region, respectively. However, the peaks in Sov−50BWO/T−ZIS shifted to higher binding energy at 532.7 eV and 531.4 eV, which resulted from the fast transport of electrons from the captured region to the interface of T−ZIS. It was further demonstrated that the S^2−^ served as an electronic bridge and facilitated priority trapped electron and transfer. In addition, the W 4f of pristine Sov−BWO exhibited two peaks at 37.5 eV and 35.3 eV, which could be assigned to W^6+^ species. Analogously, the lower shifted of Sov−50BWO/T−ZIS at 37.0 eV and 34.9 eV [33]. It was speculated that W atoms were adjacent to the oxygen vacancies-rich region, resulting in high electron concentration around vacancies [34]. These results further proved the intimate interface contact between Sov−BWO and T−ZIS [35]. Moreover, the strong electronic interaction was achieved by the S^2−^ doping and construction of electron bridge, which was conducive to the spatial separation and transport of charge in Sov−50BWO/T−ZIS Z-scheme heterojunction.

The specific surface area and pore structure of the samples were studied using N_2_ adsorption-desorption isotherms and pore size distribution curves. As shown in Appendix A, the adsorption-desorption isotherms of all samples were shown as typical IV type isotherms with H3 type adsorption hysteresis loop [36]. The hysteresis loops appeared at relative pressure P/P_0_ of about 0.8–1.0, revealing that the samples are all mesoporous structures [5]. Table 1 showed that the BET specific surface area and pore volume of pure T−ZIS are 44.55 m^2^·g^−1^ and 0.1455 cm^3^·g^−1^, which was higher than that of pristine Sov−BWO (13.98 m^2^·g^−1^ and 0.0385 cm^3^·g^−1^). The higher specific surface area and pore volume provided favorable conditions for the molecular diffusion of the reactants and the approachability of active sites. Hence, pure T−ZIS had better adsorption and decontamination capacity compared to pristine Sov−BWO. In addition, all composites (Sov−25BWO/T−ZIS, Sov−50BWO/T−ZIS, Sov−75BWO/T−ZIS and Sov−100BWO/T−ZIS) exhibited the enhanced specific surface area higher than single Sov−BWO and T−ZIS samples, which were 71.47 m^2^·g^−1^, 73.16 m^2^·g^−1^, 62.48 m^2^·g^−1^ and 50.61 m^2^·g^−1^, respectively. The self-assembly Sov−BWO and T−ZIS nanosheets probably possessed the tight heterogeneous interface contact and a puffier morphology, resulting in a larger specific surface area. With the addition of Sov−BWO, the adsorption capacity of TCH increased significantly, indicating that the construction of Sov−BWO/T−ZIS Z-scheme heterojunction was bifunctional in adsorbing and in situ decontamination to remove pollutants. Meanwhile, the Sov−50BWO/T−ZIS sample possessed the maximum specific surface area of 73.16 m^2^·g^−1^, which was responsible for the enhancement of adsorption capacity. However, the further addition of Sov−BWO resulted in a significant decrease in the specific surface area of Sov−75BWO/T−ZIS and Sov−100BWO/T−ZIS samples, which may be caused by excessive accumulation of Sov−BWO nanosheets during the self-assembly process. Appendix A exhibited the pore distribution of Sov−BWO, T−ZIS, Sov−25BWO/T−ZIS, Sov−50BWO/T−ZIS, Sov−75BWO/T−ZIS and Sov−75BWO/T−ZIS samples. The nanosheet morphology by the self-assembly process had both mesopores located within the nanosheet and mesopores with large continuous nanosheet spacing. The unique hierarchical structure exhibited multiple internal reflections and beneficial diffusion properties which enhanced the ability of photon capture. Compared with other composite materials, Sov−50BWO/T−ZIS had a smaller mesoporous particle size distribution, which was probably conducive to exposing more active sites in the composite material, accelerating the transfer rate of photogenerated carriers, thereby increasing the photocatalytic activity.

### 3.2. Adsorption and Photocatalytic Performances

As an effective broad-spectrum antibiotic with outstanding antibacterial property, tetracycline hydrochloride (TCH) has excellent antibacterial properties and has been widely used in the medical field. Herein, performance test experiments using the synthesized samples as adsorbents and photocatalysts were performed to evaluate the removal efficiency of TCH. As displayed in Figure 3a, the adsorption capacity of pure T−ZIS (37%) was significantly higher than that of Sov−BWO (14%), attributing to the large specific surface area of T−ZIS. With the increased incorporation of Sov−BWO, adsorption quantities of targeted TCH increased observably, demonstrating the superiority in adsorption by Sov−BWO/T−ZIS samples. The adsorption performance of Sov−50BWO/T−ZIS sample reached the maximum (61%), which proved that the self-assembly morphology was a feasible way to improve the adsorption capacity. Meanwhile, the enhancement of adsorption performance of Sov−50BWO/T−ZIS after self-assembly optimization was mainly attributed to the increase of specific surface area and pore volume. The removal efficiency of Sov−BWO, T−ZIS, Sov−25BWO/T−ZIS, Sov−50BWO/T−ZIS, Sov−75BWO/T−ZIS and Sov−100BWO/T−ZIS were 67%, 47%, 89%, 95%, 88% and 86%, respectively. Moreover, the Sov−50BWO/T−ZIS photocatalysts exhibited an enhanced TCH degradation rate of 93% in 15 min (Figure 3b). It was attributed to the optimized morphology, high specific surface area and rapid migration of photoexcited charge. We further analyzed the photocatalytic degradation rate under different light times to analyze the photocatalytic kinetics. The photocatalytic kinetics of T−ZIS was higher than that of Sov−BWO, which was attributed to the rapid carrier separation ability of twinned crystal T−ZIS. The Sov−50BWO/T−ZIS Z-scheme heterojunction exhibited the maximum TCH degradation rate, which was attributed to the optimized morphology and rational Z-scheme heterojunction. The doped S^2−^ served as electron bridge, accelerating the migration of photoexcited charge in Z-scheme heterojunction and enhancing photocatalytic kinetics. Figure 3c illustrated that the degradation of TCH over the as-obtained photocatalysts follows the pseudo-first-order rate constants. The rate constants of Sov−BWO, T−ZIS, Sov−25BWO/T−ZIS, Sov−50BWO/T−ZIS, Sov−75BWO/T−ZIS and Sov−100BWO/T−ZIS samples were 0.008 min^−1^, 0.0108 min^−1^, 0.0226 min^−1^, 0.03 min^−1^, 0.0209 min^−1^ and 0.0195 min^−1^, respectively, which demonstrated that the construction of Z-scheme heterojunction retained the strong redox ability of T−ZIS and Sov−BWO. Meanwhile, due to the existence of electron bridge in the Z-scheme system, the separation of photoexcited carriers was accelerated, and more O_2_^−^, h^+^· and ·OH were produced in the reaction solutions, greatly improving the photocatalytic degradation rate of TCH.

In addition, the active species scavenging experiments were carried out to identify the active species involved in the TCH removal of Sov−50BWO/T−ZIS photocatalyst. EDTA-2Na, isopropanol and p-benzoquinone are chosen as h^+^, OH and ·O_2_^−^ scavengers, respectively. As shown in Figure 3d, compared with the reaction without sacrificial agent, the addition of isopropanol had only slightly changed, indicating that OH was not the main active radicals in the reaction. After adding EDTA-2Na and p-benzoquinone, the degradation rate was greatly decreased, indicating that the active species of Sov−50BWO/T−ZIS Z-scheme heterojunction to degrade TCH were mainly h^+^ and ·O_2_^−^, which were consistent with previous literature [37]. For both pure T−ZIS and Sov−BWO samples, the addition of trapping agent exhibited a certain inhibitory effect, probably due to the consumption of active species.

Under the same conditions, five successive photocatalytic experiments were carried out on decontamination and adsorbent regeneration. Appendix A revealed that Sov−50BWO/T−ZIS sample maintained high activity after five runs. It was obviously observed that the concentration of TCH in the adsorption process remained basically unchanged, indicating that the adsorbed TCH could be effectively decontaminated through the subsequent photodegradation process. In addition, by comparing the adsorption and degradation performance after each reaction, the Sov−50BWO/T−ZIS photocatalyst realized stable adsorption and sustained degradation (Appendix A). It revealed that the degradation of TCH adsorbed onto Sov−50BWO/T−ZIS nanosheets regenerated the adsorption performance of the material. Moreover, Appendix A showed that the XRD pattern of Sov−50BWO/T−ZIS hybrid was similar before and after the reaction. Appendix A exhibited the similar characteristic peak in FT-IR spectra, indicating the Sov−50BWO/T−ZIS sample was stable.

### 3.3. Possible TCH Degradation Pathway

To identify the potential decontamination pathway of TCH during the photocatalytic stage, the intermediate products were analyzed by LC-MS systems. The corresponding m/z (mass-to-charge ratio) were presented in Appendix A. The products with m/z value of 491, 445, 433, 427,410, 352, 346, 256, 246, 242 and 184 produced during the degradation of TCH in the presence of the Sov−50BWO/T−ZIS photocatalyst. Based on previous related reports and analysis results, three possible degradation pathways were presented in Figure 4. The typical functional groups of double bond, phenolic group and amine group in TCH were possess high electron density, which were easily attacked by the active species [38,39]. For pathway 1: the transformation (m/z = 445) from TCH to intermediate product (m/z = 433) was started by dealkylation reaction under the h^+^ attack. Then, the product (m/z = 346) was resulted from the deamination reaction [39,40]. For pathway II: the intermediate product (m/z = 427) could be attributed to the loss of hydroxyl groups. The product (m/z = 410) was attributable to the loss of C–NH_2_ [41]. Pathway III: The product (m/z = 491) was formed by oxidation of double bonds, leading to the addition of hydroxyl and ketone group [38]. Then, the intermediate product further transformed into product (m/z = 433) by dehydration and product (m/z =352) via oxidation. As the photocatalytic degradation proceeding, these intermediates were further oxidized to low molecular weight organics through a series of dissociating functional groups and ring opening processes, including m/z = 257, 246, 242, 184 [39]. Finally, the above-mentioned products were mineralized into harmless inorganic substances, which completely destroyed the TCH structure.

### 3.4. Optical and Electrochemical Measurement

To study the optical absorption and bandgap structure properties, the UV-vis DRS was performed to characterize. Figure 5a displayed the UV–vis DRS curves of Sov−BWO, T−ZIS and Sov−BWO/T−ZIS with different Sov−BWO concentrations. It was clearly observed that the photoabsorption scope of the composites shifted to a longer wavelength position compared with pristine Sov−BWO, suggesting that Sov−BWO was successfully coupled with T−ZIS. The energy gap (Eg) derived from UV-vis DRS were evaluated from the plots of (ahv)^2^ versus the absorbed energy [42]. The Eg values of T−ZIS and Sov−BWO were shown in Appendix A, which were 2.41 eV and 2.82 eV, respectively. Mott–Schottky (MS) curves were executed to measure the flat-band potential and semiconductor types of T−ZIS and Sov−BWO.

As displayed in Appendix A, the MS curves of T−ZIS and Sov−BWO were positive, demonstrating the *n*-type characteristic. The flat band potential of T−ZIS and Sov−BWO were −0.71 and −0.41 V (versus Ag/AgCl, PH = 7), respectively. The flat band potential of negative 0.1 V was considered to be the conduction band edge (E_CB_) of *n*-type semiconductors [30]. Therefore, the CB potentials of T−ZIS and Sov−BWO were calculated to be −0.81 and −0.51 V (versus Ag/AgCl, PH = 7), which were equal to −0.61 and −0.31 V (versus NHE, PH = 7), respectively [43]. Meanwhile, the valence band edge (E_VB_) of T−ZIS and Sov−BWO were estimated to be 1.80 and 2.51 V versus NHE according to the formula: EVB = ECB + Eg.

To investigate the photoexcited carrier separation of as-obtained photocatalysts, PL spectra were performed at an excitation wavelength of 325 nm. As shown in Figure 5b, the PL emission intensity of the Sov−BWO/T−ZIS Z-scheme heterojunctions presented the decreased intensity compared with original Sov−BWO and T−ZIS, indicating that the probability of carrier recombination in the composite photocatalysts was effectively suppressed. It mainly caused by the formation of built-in electric field at the heterogeneous interface, which accelerated the separation and transfer of electron–hole pairs [44]. Moreover, the as-obtained Sov−50BWO/T−ZIS hybrid photocatalyst exhibited the lowest intensity, revealing that the electrons could be rapidly transferred and separated by heterogeneous interface via electron bridge path. In addition, both original Sov−BWO and Sov−50BWO/T−ZIS hybrid exhibited obvious EPR signals at a g-value of 2.003 (Appendix A), demonstrating the existence of oxygen vacancies [45]. Especially, after coupling of Sov−BWO with /T−ZIS, the EPR response was higher than that of Sov−BWO, attributing that the optimized self-assembly process of nanosheets reduced stacking of Sov−BWO and exposed more surface oxygen vacancies.

Furthermore, I-t curves and EIS measurements were employed to examine the transport behavior of photoexcited carriers. As displayed in Figure 5c, it was obviously displayed that Sov−50BWO/T−ZIS Z-scheme heterojunction had the highest photocurrent as a comparison to single Sov−BWO and T−ZIS, demonstrating the maximum separation efficiency of photoexcited carriers [33]. Similar trends have been observed in Nyquist plot (Figure 5d). Generally, the charge transfer resistance was inversely proportional to the arc diameter of Nyquist plot. Appendix A listed the Rs, Rp and CPE values of the equivalent circuit, indicating that Rp was the main resistance. An EIS Nyquist plot of the Sov−50BWO/T−ZIS photocatalyst exhibited the smallest diameter of the semicircular (1182.3 Ω) in comparison to those of the pure single component, implying the rapid separation and transport of photoexcited charge at the interface.

### 3.5. Electron Transfer Mechanism of Z-Scheme Heterojunction

To gain insight into the Z-scheme photocatalytic mechanism, we further performed in situ EPR studies to detect the responses of DMPO-·OH and DMPO-·O_2_^−^ of Sov−BWO, T−ZIS and Sov−50BWO/T−ZIS photocatalysts, respectively. Figure 6a showed that all samples had no in situ EPR signals in the dark environment, indicating that the active species were generated by photocatalysis. Under light irradiation, the DMPO-·OH results of in situ EPR detection showed that ·OH produced by Sov−50BWO/T−ZIS originated from Sov−BWO. The intensity of DMPO-·OH over the Sov−50BWO/T−ZIS was stronger than that of Sov−BWO, proving the improved photogenerated carrier separation and transmission efficiency. We had hardly detected the DMPO-·OH signal after light irradiation over pure T−ZIS. It was mainly because the EVB potential of T−ZIS was lower than that of ·OH/OH^−^ (1.99 V vs. NHE) and H_2_O/OH (2.34 V vs. NHE), resulting in the h^+^ in the CB of T−ZIS was unable to generate ·OH [22]. As displayed in Figure 6b, the obvious DMPO-·O_2_^−^ signals were discovered in T−ZIS and the DMPO-·O_2_^−^ signals could hardly be detected in Sov−BWO. The inexistence of·O_2_^−^ (O_2_/OH, −0.33 V vs NHE) over Sov−BWO was due to the inadequate E_CB_ potential [46]. Meanwhile, after constructing of Sov−50BWO/T−ZIS hybrid, the intensity of DMPO-·O_2_^−^ was significantly higher than that of T−ZIS. The coexistence of O_2_^−^ and OH species confirmed that the charge separation pathway was a Z-type heterojunction, rather than the typical II-type heterojunction [47]. These results further indicated that the Z-scheme system was conducive to the transport of photogenerated charge, thus demonstrating the robust photocatalytic activities. Based on the above active species scavenging experiments and in situ EPR results, we speculated the possible photocatalytic reactions as follows:
Sov−BWO/T−ZIS + hν → e^−^ + h^+^ (e^−^CB)(2)
T−ZIS + O_2_ → O_2_^−^ (h^+^VB)(3)
Sov−BWO + H_2_O → OH(4)
O_2_^−^ + Pollutants → degraded products(5)
H^+^ + Pollutants → degraded products(6)
OH + Pollutants → degraded products(7)

Based on the above analysis, a reasonable mechanism of adsorption and in situ decontamination in the sequential process was preliminarily confirmed between Sov−BWO and T−ZIS. As shown in Figure 6c, due to increased specific surface area and abundant mesoporous of Sov−BWO/T−ZIS, the adsorptive sites were beneficial to adsorption of more pollutants. Hence, rapid enrichment of pollutants was achieved inside or on the surface of the Sov−BWO/T−ZIS nanosheets. After adsorption equilibrium, the main active species (O_2_^−^ and h^+^) mineralized adsorbed pollutants into CO_2_ and H_2_O under visible light illumination. Herein, the electrons on the EVB of T−ZIS generated·O_2_^−^ by reducing dissolved oxygen. The holes on the EVB of Sov−BWO produced OH by oxidized H_2_O and OH^−^ [37]. Meanwhile, the photoexcited electrons on the E_CB_ of Sov−BWO were injected into the E_VB_ of T−ZIS through the Sov and rapidly transfer by S^2−^ electronic bridge, and promptly recombined with the photoexcited holes, which was consistent with the XPS analysis. Therefore, the Sov−BWO/T−ZIS Z-scheme heterojunction not only promoted the separation and transport of photoexcited carrier charges, but also retained the high redox capabilities of Sov−BWO and T−ZIS, thus exhibiting the high degradation efficiency.

## 4. Conclusions

In summary, we proposed the Z-scheme heterojunction via coupling Sov−BWO with T−ZIS nanosheets, which acted as an efficient functional photocatalyst for removal of pollutants. The addition of T−ZIS nanosheets significantly optimized the cohesion of nanosheets and improved the adsorption ability of pristine Sov−BWO for removal of tetracycline. The enhanced adsorption capacity of self-assembly Sov−BWO/T−ZIS nanosheets was attributed to the increase in specific surface area, which was favorable for the subsequent photocatalytic process. The radicals ·O_2_^−^ and h^+^ were mainly species, which were responsible for photodegradation. Therefore, the best-performing Sov−50BWO/T−ZIS sample exhibited an excellent TCH removal rate of 95%, higher than Sov−BWO (47%) and T−ZIS (67%). This study uncovers the correlation between adsorption and degradation, and provides a new insight in the design and application of direct Z-scheme heterojunction photocatalysts, which exhibits remarkable potential for the removal of pollutants.

## Data Availability

Data are contained within this article and the Appendix A.

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
