# Peer review of "Investigation of the Kinetics and Reaction Mechanism for Photodegradation Tetracycline Antibiotics over Sulfur-Doped Bi2WO6-x/ZnIn2S4 Direct Z-Scheme Heterojunction"

_nanomaterials, 2021, doi:10.3390/nano11082123_

Round 1
Reviewer 1 Report
The manuscript “Investigation of Kinetics, Reaction Mechanism for Photodegradation Tetracycline Antibiotics over Sulfur-Doped Bi2WO6- 3 x/ZnIn2S4 Direct Z-Scheme Heterojunction” represents a very interesting contribute to the field of antibiotics photodegradation.
The materials have been carefully selected to reach the goal and they were accurately characterized. The experiments are clearly described and the results explained with care. A relevant number of techniques have been used to characterize the materials and the systems and the chemical physical aspects of the photoinduced processes have been clearly explained.
Finally, the manuscript is well written and reads easily.
Some minor issues should be addressed:
- The unite of measurement of temperature has to be the same in the whole manuscript (sometimes the authors used Celsius and sometimes Kelvin). Further, please indicate Kelvin degrees with the capital letter.
- The quality of the imagines reported in figure 1 has to be strongly improved, I can’t distinguish the labels in boxes a, b and f .
- The sentence “the absorption intensity of the Sov-BWO/T-ZIS hybrids were much stronger than those of Sov-BWO and T-ZIS, indicating that the heterojunction photocatalysts could utilize visible light more effectively, thereby generating more photoexcited carriers during the photocatalysis process” should be more deeply explained or removed from the manuscript (it is not relevant for the photocatalyst efficiency).
Author Response
Reviewer: 1
1) The unite of measurement of temperature has to be the same in the whole manuscript (sometimes the authors used Celsius and sometimes Kelvin). Further, please indicate Kelvin degrees with the capital letter.
R: Thanks to the reviewer’s comment, we have carefully revised all temperature units to kelvin degrees.
2) The quality of the imagines reported in figure 1 has to be strongly improved, I can’t distinguish the labels in boxes a, b and f.
R: We have replaced the original figure 1 in revised figure caption with high resolution.
3) The sentence “the absorption intensity of the Sov-BWO/T-ZIS hybrids were much stronger than those of Sov-BWO and T-ZIS, indicating that the heterojunction photocatalysts could utilize visible light more effectively, thereby generating more photoexcited carriers during the photocatalysis process” should be more deeply explained or removed from the manuscript (it is not relevant for the photocatalyst efficiency).
R: Thanks to the reviewer’s comment, we have removed the misleading sentence in the revised manuscript.
Reviewer 2 Report
The manuscript entitled "Investigation of kinetics, reaction mechanism for photodegradation tetracycline antibiotics over sulfur-doped Bi2WO6-x/ZnIn2S4 direct Z-scheme heterojunction" (Ms. Ref. nanomaterials-1307287) reports an example of a Z-type heterojunction photocatalyst for the removal of tetracycline hydrochloride. Heterostructured photocatalysts responding in the visible-light region of the electromagnetic spectrum represent one of the latest research directions in the field of photocatalysis, and it could be attractive to a broad audience. Therefore, attention is paid to the effect of the S2- doping of Bi2WO6-x coupled with twin crystal ZnIn2S4 nanosheets, which has been investigated thoroughly using appropriately chosen complementary instrumental techniques. Testing of photocatalytic activities was done on tetracycline antibiotic substrate, which is more suitable for evaluating photocatalytic activity in the visible-light region than simple dyes as model pollutants. The Z-scheme mechanism was verified by using scavengers blocking electrons/holes effectively. The manuscript is well-balanced, the text is readable, logically structured, and flows with minimum typos, findings are supported by results and discussed adequately. Nevertheless, the manuscript should be improved according to the suggestions below:
- Minor English revisions must be done throughout the text (e. g. line 240 – Meanwhile; line 302 – 2-times significantly; some punctuation and articles before the nouns).
- Unify the unit format and control where needed. For example, "k" and "K (Calvins)." – line 112 and 113 ("k"), and line 124 there is °C.
- Line 140; Is there any difference between the electron paramagnetic resonance (EPR) and electron spin resonance (ESR)?
- Figure at line 196 should be in a Double Column format. It is hardly resolved in the current format.
- Detail description of the photocatalytic experiment should be added to the experimental section. Information about conditions and experimental arrangements are disseminated through the text or even missing. For example, the concentration of TCH can be found in supplementary characteristics of the light irradiation source and/or cut-off(on) filters used are missing as well as light irradiation intensity.
- ROS trapping experiment using scavengers is not involved for single photocatalysts components. However, it would be interesting to investigate the prevailing mechanism (i.e., via electrons or holes) for samples T-ZIS and sov-BWO.
- Line 278-279; the sentence "It was revealed that self-assembly possess was a viable strategy to improve the specific surface area" does not give good meaning; please modify it. The influence of specific surface area on photocatalysts performance is not straightforward, see for example:
https://doi.org/10.1016/j.apsusc.2020.145773
https://doi.org/10.1039/C2CS35378D
- Line 399-401; estimation of valence bands positions should be done with precaution. Consider pH dependence (values differ significantly for pH0 and 7), or at least indicate which pH these values hold. Information about the pH of the system in solution, before and after the photocatalytic experiment, can be helpful for the estimation in this paper. The same hold for values given in Fig. 6c. For more information on this issue, see:
https://doi.org/10.3390/catal11020293
https://doi.org/10.2138/am-2000-0416
- The presentation of results on testing of photocatalytic performance of photocatalysts is somewhat misleading. In Figure 3a, the x-axis does not represent a rate (btw., unit is missing), rather a degree of conversion. It seems from data presented in Fig. 3a that 30 min is not enough for reaching adsorption/desorption equilibrium as indicated by steep slope after switch on light followed by level-off
- Photocorrosion of photocatalysts is often encountered during exposure to light in aqueous environments, especially for sulfide materials. Even though the design of photocatalysts reusability test is not simple and may be burdened with errors, it is desirable to at least exclude structural changes in photocatalysts by XRD, see for example;
https://doi.org/10.3390/catal11020293
https://doi.org/10.1016/j.matchemphys.2019.121823

Author Response
Reviewer: 2
1) Minor English revisions must be done throughout the text (e. g. line 240 – Meanwhile; line 302 – 2-times significantly; some punctuation and articles before the nouns).
R: Thanks to the reviewer’s comment, we have checked the manuscript and corrected the writing and grammar errors carefully.
2) Unify the unit format and control where needed. For example, "k" and "K (Calvins)." – line 112 and 113 ("k"), and line 124 there is °C.
R: We have carefully revised units to kelvin degrees in the revised manuscript.
3) Line 140; Is there any difference between the electron paramagnetic resonance (EPR) and electron spin resonance (ESR)?
R: Thanks to the reviewer’s comment, we have corrected the writing made of ESR to EPR.
4) Figure at line 196 should be in a Double Column format. It is hardly resolved in the current format.
R: Considering the reviewer’s suggestion, we have revised the Figure 2 images showed in below figure.
5) Detail description of the photocatalytic experiment should be added to the experimental section. Information about conditions and experimental arrangements are disseminated through the text or even missing. For example, the concentration of TCH can be found in supplementary characteristics of the light irradiation source and/or cut-off(on) filters used are missing as well as light irradiation intensity.
R: Thanks to the reviewer’s comment, the involved detailed description of the photocatalytic experiment has been added to the experiment section.
6) ROS trapping experiment using scavengers is not involved for single photocatalysts components. However, it would be interesting to investigate the prevailing mechanism (i.e., via electrons or holes) for samples T-ZIS and sov-BWO.
R: Thanks to the reviewer’s comment, We have done related experiments, as shown in the figure below:
7) Line 278-279; the sentence "It was revealed that self-assembly possess was a viable strategy to improve the specific surface area" does not give good meaning; please modify it. The influence of specific surface area on photocatalysts performance is not straightforward, see for example:
https://doi.org/10.1016/j.apsusc.2020.145773
https://doi.org/10.1039/C2CS35378D.
R: Thanks for the reviewer's correction, the sentence "It was revealed that self-assembly possess was a viable strategy to improve the specific surface area" was incomplete. It revealed that the self-assembly process of the nanosheet promoted the closer contact between the two components, resulting in a puffier morphology of the nanosheet and a larger specific surface area (https://doi.org/10.1016/j.apcatb.2019.117997). Therefore, we speculated believed that the self-assembly process was a favorable synthesis strategy. According to the reviewer's comment, the description of the effect of surface area on the performance was also changed. (https://doi.org/10.1016/j.apsusc.2020.145773,
https://doi.org/10.1039/C2CS35378D).
8) Line 399-401; estimation of valence bands positions should be done with precaution. Consider pH dependence (values differ significantly for pH0 and 7), or at least indicate which pH these values hold. Information about the pH of the system in solution, before and after the photocatalytic experiment, can be helpful for the estimation in this paper. The same hold for values given in Fig. 6c. For more information on this issue, see:
https://doi.org/10.3390/catal11020293
https://doi.org/10.2138/am-2000-0416
R: Thanks to the reviewer’s comment, we supplemented the value for pH. In a limited time, we tested the initial PH of TCH at 6.4, which remained stable at about 7 after the addition of samples.
9) The presentation of results on testing of photocatalytic performance of photocatalysts is somewhat misleading. In Figure 3a, the x-axis does not represent a rate, rather a degree of conversion. It seems from data presented in Fig. 3a that 30 min is not enough for reaching adsorption/desorption equilibrium as indicated by steep slope after switch on light followed by level-off.
R: Thanks to the reviewer’s comment, we have corrected the misleading writing carefully in Figure 3a. Considering the reviewer’s suggestion and in order to achieve adsorption/desorption equilibrium, we also tested the adsorption removal rate of Sov-50BWO/T-ZIS under dark conditions as shown in Figure S9. According to Figure S9, the removal of 30 minutes by dark adsorption has reached 60%, and that of 90 minutes by dark adsorption is little higher of 63%. Therefore, we think that 30 minutes has basically reached the adsorption/desorption equilibrium. As illustrated in following figure:
10) Photocorrosion of photocatalysts is often encountered during exposure to light in aqueous environments, especially for sulfide materials. Even though the design of photocatalysts reusability test is not simple and may be burdened with errors, it is desirable to at least exclude structural changes in photocatalysts by XRD, see for example;
https://doi.org/10.3390/catal11020293
https://doi.org/10.1016/j.matchemphys.2019.121823
R: Thanks to the reviewer’s comment, photocorrosion of sulfide materials seriously affects the performance of photocatalysis. The cyclic experiment and the XRD and FT-IR results were employed to reveal the stability, which was displayed in Figure S6. The relevant statements about cyclic experiment are in the line 357-368 (https://doi.org/10.3390/catal11020293,https://doi.org/10.1016/j.matchemphys.2019.121823).
